# Synthesis of Carbosilane and Carbosilane-Siloxane Dendrons Based on Limonene

**DOI:** 10.3390/polym14163279

**Published:** 2022-08-12

**Authors:** Aleksei I. Ryzhkov, Fedor V. Drozdov, Georgij V. Cherkaev, Aziz M. Muzafarov

**Affiliations:** 1A. N. Nesmeyanov Institute of Organoelement Compounds, Russian Academy of Sciences, 119991 Moscow, Russia; 2Enikolopov Institute of Synthetic Polymeric Materials, Russian Academy of Sciences, 117393 Moscow, Russia

**Keywords:** limonene, carbosilane dendrimers, siloxanes, hydrosilylation

## Abstract

In this work, carbosilane dendrons of the first, second, and third generations were obtained on the basis of a natural terpenoid, limonene. Previously, we have shown the possibility of selective hydrosilylation and hydrothiolation of limonene. It is proved that during hydrosilylation, only the isoprenyl double bond reacts, while the cyclohexene double bond does not undergo into the hydrosilylation reaction. However, the cyclohexene double bond reacts by hydrothiolation. This selectivity makes it possible to use limonene as a dendron growth center, while maintaining a useful function—a double bond at the focal point. Thus, the sequence of hydrosilylation and Grignard reactions based on limonene formed carbosilane dendrons. After that, the end groups were blocked by heptamethyltrisiloxane or butyllithium. The obtained substances were characterized using NMR spectroscopy, elemental analysis and GPC. Thus, the proposed methodology for the synthesis of carbosilane dendrons based on the natural terpenoid limonene opens up wide possibilities for obtaining various macromolecules: dendrimers, Janus dendrimers, dendronized polymers, and macroinitiators.

## 1. Introduction

Dendrimers, due to the particular properties of their structure, which determines the set of unique properties of these molecules, have found a wide range of applications since their first preparation in: catalysis, materials science, and nanobiotechnology [1,2,3,4,5]. At present, interest in the field of such molecules is determined by their well-defined, regular and homogeneous branched structure, the presence of a large number of functional groups on the periphery of the molecule, and a wide range of typologies [6,7,8].

One of the promising classes of dendrimeric molecules are carbosilane dendrimers formed by a carbon–silicon skeleton [9,10,11,12]. The high chemical and thermal stability of these dendrimers is associated with the high energy of the C-Si bond, as well as its low polarity. The last characteristic of the C–Si bond provides the high hydrophobicity of the carbosilane dendrimers. Nevertheless, they can be easily modified by functionalizing the periphery with polar moieties, turning them into hydrophilic molecules and thus, carbosilane dendrimers can also be used in biomedical applications such as a delivery systems of gene materials, drugs, etc. [13,14,15,16,17,18,19,20].

A great breakthrough was done in the synthesis and study of new self-organizing nanostructured objects—dendrimersomes. Dendrimersomes, by analogy with polymersomes, can be classified as self-organizing objects consisting of amphiphilic dendrimeric structures, which can be constructed with Janus dendrimers of different types. The Virgil Percec’s group synthesized an extensive library of Janus dendrimers based on the natural component—gallic acid [21].

Further studies showed that the different types of Janus dendrimers can spontaneously form nanosized objects of various shapes: vesicles, dendrimerosomes, cubosomes, disks, domed vesicles, and helical structures. It was also shown that the obtained objects are stable over a long time, have better mechanical characteristics compared to polymerosomes, and are more uniform in particle size. In addition, dendrimerosomes are easier to form than polymerosomes and are easier to functionalize.

The main question faced by researchers of dendrimerosomes was the question about the origins for one or another type of spontaneous structuring. How does the geometry of a macromolecule and its structure affect the type of nanosized objects formed? To answer this question, many works were done, including computer modeling, in which various types of intermolecular interactions were discussed: ion–ion, donor–acceptor, hydrogen bonds, ion–dipole interactions, π-stacking, and their influence on the final structure of the nanoscale objects [22,23,24,25,26,27].

Nevertheless, most of the obtained nanosized dendrimeric objects were characterized phenomenologically using physicochemical methods of analysis. Since the majority of dendritic structures obtained on the basis of Percec-type monomers (benzylphenylene ether dendrons) have a mesogenic character, it was possible to determine the type of ordering of such objects by determining the type of LC phase. In addition, with the development of modeling methods and X-ray diffraction analysis, it became possible to create simulations of self-assembling dendrimer structures depending on the structure of dendrons [28], which made it possible to further describe real dendrimersomes, including Percec-type objects, using this approach. Thus, in recent works, it was shown that a change in the chemical structure of the focal group, repeating unit, and peripheral groups in dendrons leads to a change in the type of structure of the final nano-object [29]. Thus, it can be concluded that, by varying the chemical structure of dendrons or dendrimers, it is possible to change the self-organization and ordering of macromolecules, which leads to the creation of a molecular design paradigm, molecular containers with programmed properties, and other principles of nanotechnology. Nowadays, a wide variety of Janus dendrimers have been obtained using various modern chemical approaches and starting compounds. Based on polyimines with polysulfides, amphiphilic Janus dendrimers were obtained by convergent synthesis [30]. Light-sensitive Janus dendrimers with a hyperbranched structure were obtained based on poly(3-ethyl-3-oxetanemethanol) [31]. The so-called glycodendrimerosomes, dendrimers with functional glycosidic groups on the periphery, have become widespread [32]. Dendrimers with functional glycosidic groups can also form cubosomes, lamellae, and micelles [33]. In recent years, Percec’s group has investigated the co-association of dendrimers with functional glycosidic groups with the walls of *E. Coli* bacteria. For this purposes, amphiphilic dendrons containing a long alkyl fragment and a dendritic polyamide structure were synthesized. The binding of the two fragments was carried out using click chemistry [34].

It should be noted that despite the growing interest in organosilicon dendrimers, caused by their model nature, which in turn is determined by the chemical inertness of the carbosilane skeleton, these objects are systematically studied by only a few scientific groups. Therefore, it is not surprising that there are very few works concerning the synthesis and study of the properties of organosilicon Janus dendrimers. In [35], the authors showed a simple functionalization of a carbosilane dendron using the Staudinger reaction. However, the work is purely synthetic. One can also cite an article by the same group [36], although the resulting dendrimers can be called Janus dendrimers only formally. In addition, the obtained structures contain only triethoxysilyl groups, which cannot determine the structure and features of organosilicon dendrimers.

Concerning the synthesis of dendrimers, there are two typical schemes—divergent and convergent approaches. According to the divergent approach, the growth of the dendrimer starts from the *X_n_* branching center (starting branching center) by sequentially attaching elementary units with different functionality *YZ_n_*, while the *X* functions can form a chemical bond with *Y*, but not with *Z*. In addition, *Y* does not can react with *Z*. A fundamentally different approach is called convergent. According to this scheme, the synthesis of a monodendron is first carried out—a molecule that can be imagined as a segment of a dendrimer symmetrical about its center—a focal point. Thus, at the first stage, the synthesis of a monodendron is carried out, and at the second stage, the formation of a dendrimer occurs by attaching various monodendrons to a branching center with a functionality of *n*. The direction of organosilicon dendrimers, including carbosilane ones, is being actively developed by Muzafarov’s group [37,38] and being studied in parallel by several groups, including Frey’s group [39,40]. Further, a large number of organosilicon dendrimers with different chemical structures and architectures were obtained using one of these approaches or their combination.

The aim of this work was to develop a simple and convenient method for the synthesis of carbosilane dendrons with a hidden functional group at the focal point. This makes it possible to develop the structure of the obtained molecules both according to the divergent and convergent schemes. We also expect to use them in the synthesis of Janus dendrimers in the future. An important factor is also that the obtained dendrons are bioinert and biocompatible due to the nature of their carbosilane structure and the use of a natural terpene—limonene.

The chemistry and technology of limonene and its derivatives is well studied and described in the literature [41,42]. In addition, the hydrosilylation reaction of limonene has previously been described in the literature. Thus, the first mention of the hydrosilylation of limonene with dichloromethylsilane in the presence of a Speier’s catalyst was in 1968 [43], and later in subsequent publications [44,45]. In addition, the hydrosilylation reaction of limonene is described in the classic handbook on the hydrosilylation reaction by Marciniec [46] and described in his latest works [47].

## 2. Material and Methods

All reactions were carried out under an inert atmosphere, and solvents were purified from appropriate drying agents. D-limonene (Sigma-Aldrich, The Netherland), allylchloride (ABCR, Germany), dichloromethylsilane (ABCR, Germany), n-butyllithium (solution 2.5 M in hexane) (Sigma-Aldrich, The Netherland), hexachloroplatinic(IV) acid hexahydrate (Speier’s catalyst) (Sigma-Aldrich, The Netherland), and Karstedt’s Pt catalyst (2% solution in o-xylene) (ABCR, Germany) were obtained from commercial sources.

Gel Permeation Chromatography (GPC) analysis was performed on a Shimadzu LC-10A series chromatograph (Japan) equipped with an RID-10A refractometer and SPD-M10A diode matrix detectors. For analytical separation, Phenomenex column (USA) with a size of 7.8 mm × 300 mm filled with the Phenogel sorbent with a pour size of 500 Å was used.

The GLC analysis was performed on a “Chromatech Analytic 5000” chromatograph (Russia) with katharometer as detector, helium as carrier gas, with 2 m × 3 mm column, and stationary phase SE-30 (5%) was applied to Chromaton-H-AW. Registration and data collection were carried out with the help of the program “Chromatech Analyst” (Yoshkar-Ola, Russia).

1H, 13C, 29Si Nuclear Magnetic Resonance (NMR) spectra, and their nucleus correlations were recorded using a Bruker Avance II 300 spectrometer at 300, 75, and 60 MHz, respectively.

### 2.1. Dichloro(2-(4-methylcyclohex-3-en-1-yl)propyl)methylsilane (Lim-G_0_Cl^2^)

To a stirred mixture of 20.0 g (0.15 mol) of limonene and 140 μL of Speier’s catalyst (0.067 mg/μL) we added methyldichlorosilane (25.33 g, 0.22 mol). Reaction mixture was stirred at room temperature for 24 h. The completeness of the process was controlled by ^1^H NMR spectroscopy from disappearance of proton signals at double bonds in isoprenyl groups. The reaction mixture was evaporated to remove excess of dimethylchlorosilane.

### 2.2. Diallyl(2-(4-methylcyclohex-3-en-1-yl)propyl)methylsilane (Lim-G_0_All^2^)

To a stirred suspension of magnesium chips (18.27 g, 0.75 mol) in anhydrous THF (40 mL) we added a solution of the mixture of allyl chloride (35.95 g, 0.47 mol) and dichloromethylsilyllimonene 37.65 g (0.15 mol) in anhydrous THF (130 mL). Owing to the heat effect, the process was carried out at the boiling temperature of the mixture of solvents (60–65 °C). When addition of the mixture was completed, the reaction solution was stirred at 65 °C for 3.5 h. The excessive amount of allylmagnesium chloride was deactivated with the saturated aqueous solution of NH_4_Cl. The precipitate of MgCl_2_ was filtered off and washed with n-hexane on the filter. The filtrate was evaporated to remove the solvents, and the residue was evacuated (0.5 mbar) at 60 °C. After distillation (0.1 mbar), 50 g of final product was obtained. B.p. = 110–112 °C/0.5 mbar. Yield of the product was 82%. Calculated for C_17_H_30_Si: C 77.71%; H 11.43%; Si 10.67%. Found: C 77.39%; H 11.25%; Si 10.31%.

### 2.3. Trichloro(2-(4-methylcyclohex-3-en-1-yl)propyl)silane (Lim-G_0_Cl^3^)

This compound was obtained similarly to *dichloromethylsilyllimonene* from 10.0 g (0.073 mol) limonene, 90 μL of Speier’s catalyst (0.067 mg/μL), and 12.93 g (0.095 mol) trichlorosilane.

### 2.4. Triallyl(2-(4-methylcyclohex-3-en-1-yl)propyl)silane (Lim-G_0_All^3^)

This compound was obtained similarly to *diallylmethylsilyllimonene* from 19.83 g (0.073 mol) *trichlorosilyllimonene*, 11.13 g (0.46 mol) magnesium chips and 21.91 g (0.29 mol) allyl chloride. Yield of the product was 85%. Calculated for C_19_H_32_Si: C 79.02%; H 11.09%; Si 9.70%. Found: C 75.57%; H 10.61%; Si 9.23%.

### 2.5. Dendrimer of the First Generation (Lim-G_1_Cl^4^)

To the stirred solution of *diallylmethylsilyllimonene* (2.0 g, 0.0076 mol) and 23 μL of Karsted’s catalyst in anhydrous toluene (46 mL) we added a methyldichlorosilane (2.63 g, 0.023 mol). Reaction mixture was stirred at room temperature for 24 h. The completeness of the process was controlled by ^1^H NMR spectroscopy from disappearance of proton signals at double bonds in allyl groups. The reaction mixture was evaporated to remove excess of dimethylchlorosilane.

### 2.6. Dendrimer of the First Generation (Lim-G_1_All^4^)

To a stirred suspension of magnesium chips (1.54 g, 0.063 mol) in anhydrous THF (10 mL) we added a solution of the mixture of 3.03 g (0.040 mol) allyl chloride and *Lim-G_1_Cl^4^* (3.74 g, 0.0076 mol) in anhydrous THF (20 mL). Owing to the heat effect, the process was carried out at the boiling temperature of the mixture of solvents (60–65 °C). When addition of the mixture was completed, the reaction solution was stirred at 65 °C for 3.5 h. The excessive amount of allylmagnesium chloride was deactivated with the saturated aqueous solution of NH_4_Cl. The precipitate of MgCl_2_ was filtered off and washed with n-hexane on the filter. The filtrate was evaporated to remove the solvents, and the residue was dissolved in n-hexane and passed through silica. Solvent was removed by evaporation under reduced pressure. Yield of the product was 7.5 g (66%). Calculated for C_31_H_58_Si_3_: C 72.23%; H 11.26%; Si 16.31%. Found: C 72.17%; H 11.16%; Si 16.24%.

### 2.7. Dendrimer of the Second Generation (Lim-G_2_Cl^8^)

This compound was obtained similarly from 2.0 g (0.0039 mol) *Lim-G_1_All^4^*, 2.68 g (0.023 mol) methyldichlorosilane, 23 μL of Karsted’s catalyst, and 46 mL anhydrous toluene. 

### 2.8. Dendrimer of the Second Generation (Lim-G_2_All^8^)

This compound was obtained similarly from 5.12 g (0.0039 mol) *Lim-G_2_Cl^8^*, 1.57 g (0.065 mol) magnesium chips, and 3.09 g (0.040 mol) allyl chloride. Yield of the product was 72%. Calculated for C_59_H_114_Si_7_: C 69.40%; H 11.18%; Si 19.21%. Found: C 67.23%; H 10.69%; Si 17.64%.

### 2.9. 1,1,3,5,5,5-Heptamethyltrisiloxane (HMS)

This compound was synthesized according to method published in [48]. Yield of HMS was 78% as colorless dense liquid. B.p. 140–141 °C. ^1^H NMR (300 MHz, Chloroform-d) δ: 4.65 (s, 1H, SiH), 0.12 (s, 21H, SiCH_3_).

### 2.10. 1,1,3,5,5,5-Heptamethyl-3-(2-(4-methylcyclohex-3-en-1-yl)propyl)trisiloxane (Lim-G_0,5_TMS^2^)

To a stirred mixture of 1.96 g (0.0088 mol) of HMS and 1.0 g (0.0073 mol) of limonene, purged with argon, we added 17 mL dry toluene and 17 μL of Karsted’s catalyst. Reaction mixture was stirred at 60 °C for 4 h. After distillation (0.1 mbar), 5 g of final product was obtained. T_b_ = 115–118 °C/0.5 mbar. Yield of the product was 89%. Calculated for C_17_H_38_O_2_Si_3_: C 56.87%; H 10.59%; O 8.92%; Si 23.42%. Found: C 54.36%; H 10.01%; Si 21.00%.

### 2.11. Bis(heptamethylsilylpropyl)methylsilyllimonene (Lim-G_1,5_TMS^4^)

To a stirred mixture of 1.87 g (0.0084 mol) of HMS and 1.0 g (0.0038 mol) of *diallylmethylsilyllimonene*, purged with argon, we added 17 mL dry toluene and 17 μL of Karsted’s catalyst. Reaction mixture was stirred at 60 °C for 10 h. After reaction completed, substance was evaporated to remove the solvents, and the residue was evacuated (0.5 mbar) at 70 °C. Yield of the product was 99%. Calculated for C_31_H_74_O_4_Si_7_: C 52.58%; H 10.46%; O 9.05%; Si 27.70%. Found: C 50.02%; H 9.84%; Si 26.56%.

### 2.12. Tris(heptamethylsilylpropyl)silyllimonene (Lim-G_1,5_TMS^6^)

This compound was obtained similarly to *bis(heptamethylsilylpropyl)methylsilyllimonene* from 1.0 g (0.0035 mol) *triallylsilyllimonene*, 2.78 g (0.012 mol) HMS, 22 μL of Karsted’s catalyst, and 22 mL anhydrous toluene. Yield of the product was 99%. Calculated for C_40_H_98_O_6_Si_10_: C 50.21%; H 10.25%; O 10.04%; Si 29.29%. Found: C 48.96%; H 10.14%; Si 28.44%.

### 2.13. Dendrimer of the 2,5 Generation (Lim-G_2,5_TMS^8^)

This compound was obtained similarly to *bis(heptamethylsilylpropyl)methylsilyllimonene* from 1.0 g (0.0019 mol) ***G2(All)_4_***, 2.07 g (0.0093 mol) HMS, 12 μL of Karsted’s catalyst, and 12 mL anhydrous toluene. Yield of the product was 99%. Calculated for C_59_H_146_O_8_Si_15_: C 50.39%; H 10.39%; O 9.11%; Si 29.89%. Found: C 48.91%; H 9.95%; Si 28.27%.

### 2.14. Dendrimer of the 3,5 Generation (Lim-G_3,5_TMS^18^)

This compound was obtained similarly to *bis(heptamethylsilylpropyl)methylsilyllimonene* from 1.0 g (0.00098 mol) *Lim-G_1_All^4^*, 2.09 g (0.0094 mol) HMS, 18 μL of Karsted’s catalyst, and 18 mL anhydrous toluene. Yield of the product was 99%. Calculated for C_115_H_290_O_16_Si_31_: C 49.28%; H 10.36%; O 9.14%; Si 31.00%. Found: C 49.39%; H 10.18%; Si 29.96%.

### 2.15. Dibuthyl(2-(4-methylcyclohex-3-en-1-yl)propyl)methylsilane (Lim-G_0_Bu^2^)

To a stirred mixture of 35 mL (2.5 M) of n-BuLi in anhydrous hexane (50 mL) we added a solution of 9.25 g (0.037 mol) Lim-G_0_Cl^2^ in anhydrous hexane (50 mL) at −70 °C. When addition of the mixture was complete, anhydrous THF (70 mL) was added to the reaction solution. Reaction mixture was stirred at −70 °C, and then at room temperature for 24 h. The excessive amount of n-BuLi was deactivated with the ethanol. The precipitate of LiCl was filtered off and washed with n-hexane on the filter. The filtrate was evaporated to remove the solvents, and the residue was evacuated (0.5 mbar) at 60 °C. After distillation (0.1 mbar), 8 g of final product was obtained. B.p. = 150–153 °C/0.5 mbar. Yield of the product was 83%. Calculated for C_19_H_38_Si: C 77.40%; H 12.90%; Si 9.50%. Found: C 77.26%; H 12.94%; Si 9.27%.

## 3. Results and Discussion

By their structure, dendrimers can be considered as hierarchical macromolecules with an iterative structure. Thus, the dendrimer molecule is symmetrical about the branching center from which the branches start to grow. Further, each of the branches derives into the next generation. Theoretically, the growth and branching of the dendrimer can continue indefinitely, however, based on considerations of the van der Waals repulsion and the packing density of branches at high generations, this becomes impossible.

In this work, a convergent synthesis strategy was implemented for the synthesis of carbosilane dendrons with limonene at the focal point. This strategy has become prevalent for the preparation of cabosilane dendrons for several reasons: ease of synthesis, available starting compounds, scalability, and versatility. The synthesis of similar dendrons, containing, as an example, a focal point in the form of a chloropropyl group, is shown in Figure 1.

Briefly, the synthesis of carbosilane dendrons consists of two successive reactions, the hydrosilylation reaction and the Grignard reaction. Any compound containing more than one double bond can be taken as a starting point. The density of the branches in the final dendron will depend on the number of double bonds in the original compound (branch point). Thus, the scheme shows the growth of a carbosilane dendron starting from chloropropyl triallylsilane (ClPr-G_0_All^3^), which can be considered as a zero-generation dendron in the formalism of dendrimer chemistry. At the first stage, hydrosilylation with methyldichlorosilane is carried out, which gives the dendron of the first-generation ClPr-G_1_Cl^6^. Its treatment with an excess of Grignard reagent from allyl chloride leads to the first-generation dendron with allyl groups on the periphery (ClPr-G_1_All^6^). This dendron is completely similar to the initial chloropropyl triallyl silane (ClPr-G_0_All^3^), but differs from it in the generation number and, accordingly, in the number of allyl groups. Further, by a combination of these two reactions, dendrons of higher generations can be obtained. It should be noted that the periphery of chlorosilyl dendrons can be represented by any alkyl, alkenyl, alkynyl, and other functions that can be introduced by replacing chlorine atoms with the corresponding organometallic compounds. The scheme shows a variant of the introduction of butyl substituents (ClPr-G_1_Bu^6^) by blocking chlorosilyl groups with *n*-butyllithium.

In this work, it was shown that carbosilane dendrons can be obtained in a similar way, starting from the natural terpene, limonene. Limonene is a monoterpene terpenoid containing two double bonds in its structure: isoprenyl and cyclohexenic. Recently [49], we have shown that these double bonds have different hydrosilylation reactivities, which makes limonene an attractive compound to use as a starting compound. Thus, in the hydrosilylation reaction, the addition of the corresponding silane occurs regioselectively with the isoprenyl function, while the cyclohexene function remains unaffected. Therefore, after hydrosilylation, the cyclohexene double bond can be used as a hidden function in further transformations. It should be noted that limonene is found in large quantities in citrus fruits (up to 90%) and can be easily isolated by various methods, and, as a result, it is commercially available. For example, limonene can be used as a solvent instead of hexane and cyclohexane. In addition, limonene is chemically harmless, bioinert, and does not pose a danger to the environment.

Thus, to obtain carbosilane dendrimers with three or two arms, limonene was taken as the starting compound and hydrosilylation was carried out with various hydride silanes: trichlorosilane or methyldichlorosilane. Carrying out the hydrosilylation reaction between limonene and methyldichlorosilane catalyzed by Karsted’s catalyst resulted in a limonene conversion of only 26% in 48 h. After sequential optimizations, conditions were chosen to give a limonene conversion of more than 95%: carrying out the reaction without a solvent in the presence of a Speier’s catalyst led to the desired product Lim-G_0_Cl^2^. Similarly, the hydrosilylation reaction between limonene and trichlorosilane was carried out, which made it possible to obtain the desired product Lim-G_0_Cl^3^ in a 95% yield. Both compounds were obtained in an individual state by vacuum fractional distillation.

At the next stage, the addition of the Grignard reagent, allylmagnesium chloride, to methyldichlorosilyl limonene was carried out. The reaction proceeded at room temperature, while the progress of the reactions was monitored by ^1^H NMR spectroscopy (Appendix A). The product Lim-G_0_All^2^ was obtained in an 87% yield by vacuum distillation. Similarly, the product Lim-G_0_All^3^, based on trichlorosilyl limonene, was obtained in an 85% yield.

The subsequent growth of generations on both the two-arm Lim-G_0_All^2^ and three-arm Lim-G_0_All^3^ dendrons was completely the same as the standard procedures for the carbosilane dendrons (Figure 2, top frame).

Thus, starting from the formal nomenclature of dendrimers, we can conclude that in the course of this work, zero-generation dendrons with two Lim-G_0_All^2^ and three Lim-G_0_All^3^ allyl groups were obtained. Further, based on them, higher generations were obtained—first Lim-G_1_All^4^ and second Lim-G_2_All^8^ generations. The second part of this work devoted to show the possibility of the functionalization of the allyl periphery of dendrons. For this purposes, it was decided to proceed with the reaction of the hydrosilylation of the terminal allyl groups of dendrons with heptamethyltrisiloxane (HMS). Such a modification of dendrons is interesting, on the one hand, from the point of view of the completeness of hydrosilylation on the model’s individual HMS compound, and, on the other hand, from the point of view of sealing the peripheral shell of the carbosilane dendrons by introducing branched siloxane branches. Similar dendrimers are described in the literature. For example, in [50], carbosilane dendrimers of various generations with heptamethyltrisiloxane substituents in the shell were synthesized. A comparative analysis of the obtained structures with butyl analogues was carried out. It was shown that replacing the butyl shell with heptamethyltrisiloxane does not significantly change the nature of the flow curve, but increases the glass transition temperature of the obtained dendrimers, which is a very useful property of the obtained dendrimers. All the synthesized dendrones were investigated by ^1^H NMR, GC and GPC (Appendix A).

Thus, obtaining such systems is a fundamental task. However, you can also get an application orientation. For example, trisiloxane end groups increase gas permeability, this may be a useful property in obtaining gas permeable membranes. Additionally, trisiloxane tails create a denser shell compared to butyl analogues. This property can be realized in drug delivery systems.

The hydrosilylation reaction was carried out in excess HMS with the addition of Karsted’s catalyst. The course of the reaction was monitored by ^1^H NMR. Excess HMS was then distilled off from the reaction mixture under reduced pressure. All HMS dendrons (Figure 2, bottom frame) were obtained in high yields. In terms of the formal nomenclature of dendrimers, in the case of dendrons with HMS, the HMS residue can be considered as 0.5 of a generation due to the fact that the addition occurred at the middle silicon atom, while the two -O(CH_3_)_3_ groups attached to it can be considered as branching. However, this branching occurred in one step of HMS attachment. Thus, the product of the addition of HMS to limonene (Lim-G_0_._5_TMS^2^) can be considered as a half generation with two trimethylsiloxy (TMS) groups. Therefore, starting from dendrons with allyl functionalities on the periphery, the corresponding HMS dendrons Lim-G_1.5_TMS^4^, Lim-G_1.5_TMS^6^, Lim-G_2.5_TMS^8^ and Lim-G_3.5_TMS^18^ were obtained by hydrosilylation with HMS.

Thus, it was shown that the standard scheme for the synthesis of carbosilane dendrons can be completely transferred for obtaining analogs based on limonene. The advantage of this new synthesis methodology is the availability, bioinertness, and chemical safety of limonene as the initial natural monomer and the possibility of obtaining dendrons with an intact cyclohexene double bond as a hidden function, which is necessary when using these dendrons to obtain various macromolecules: dendrimers, Janus dendrimers, dendronized polymers, macroinitiators.

## 4. Conclusions

In this work, the strategy of the synthesis of the carbosilane dendrons based on the natural terpene limonene was demonstrated. It was shown that, using the different reactivity of two double bonds of limonene in the hydrosilylation reaction, it is possible to selectively carry out the growth of dendrons in a convergent manner, repeating the hydrosilylation reactions and the addition of allylmagnesium chloride. It was also demonstrated that the periphery of the obtained carbosilane dendrons can be changed by blocking the chlorosilane dendrons with various organometallic compounds or by a hydrosilylation reaction of terminal double bonds. The structures of all compounds were fully characterized by NMR, GPC, and elemental analysis. Thus, a simple and efficient method for the preparation of carbosilane dendrons based on limonene opens up new prospects for the preparation and functionalization of various macromolecular structures: dendrimers, Janus dendrimers, and dendronized polymers.

## Figures and Tables

**Figure 1 polymers-14-03279-f001:**
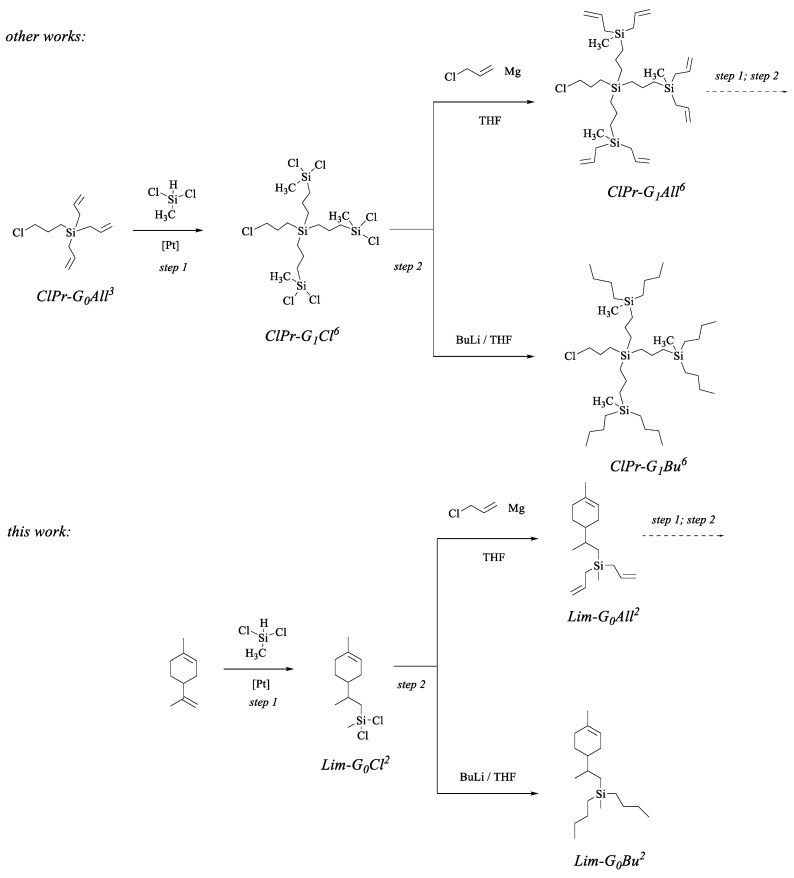
Typical scheme for the synthesis of carbosilane dendrons using the starting chloropropyl triallyl silane as an example (**top**); scheme for the synthesis of carbosilane dendrons based on limonene (**bottom**).

**Figure 2 polymers-14-03279-f002:**
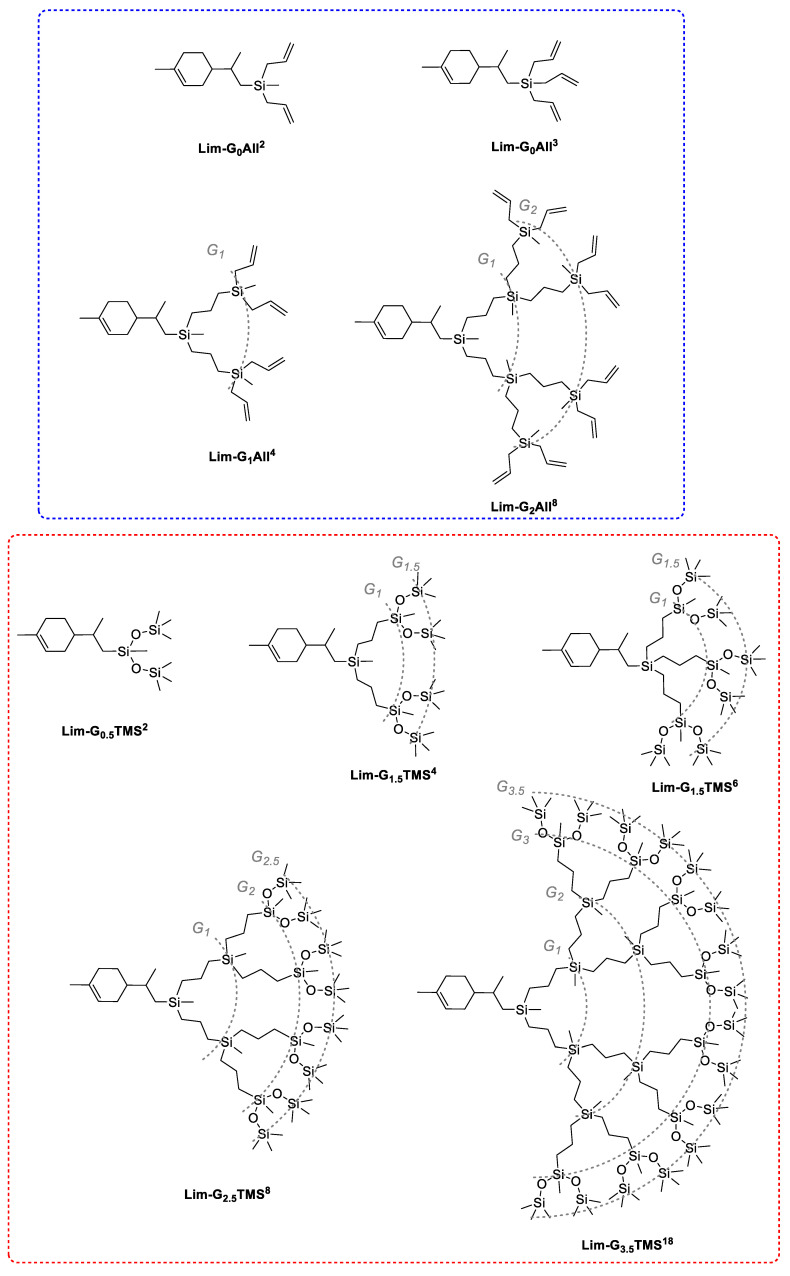
Library of obtained dendrons based on limonene with allyl functionality on the periphery and blocked with heptamethyltrisiloxane.

## Data Availability

The data presented in this study are available on request from the corresponding author.

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
