# Peer review of "Synthesis of Carbosilane and Carbosilane-Siloxane Dendrons Based on Limonene"

_polymers, 2022, doi:10.3390/polym14163279_

Round 1
Reviewer 1 Report
Muzafarov et al. described an efficient method, successive hydrosilylation and Grignard reaction, for the preparation of the first, second and third generation of new carbosilane dendrons based on limonene, the most common terpene found in nature. As the hydrosilylation is selective and occurred only on the isopropenyl double bond of limonene, it is possible to maintain a double bond at the focal point. The extended chains with allyl groups can be ended by introducing siloxane groups via quantitative hydrosilylation. These new carbosilane dendrons based on limonene obtained in this work are interesting and promising for the preparation of versatile macromolecules.
Considering the interesting results shown in this work, the acceptance is recommended. Nevertheless, there are several issues need to be improved before the acceptance.
1. A recent review concerning Janus amphiphilic dendrimers (Eur. J. Pharm. Sci., 2017, 97, 113) is worth being cited at the beginning of the Introduction.
2. In this work, the dendron growth center limonene is D-limonene. May I understand that during the synthesis of carbosilane dendrons, the stereo configuration was well retained?
3. Page 7, in the last paragraph, “Such a modification of dendrons is interesting, ……, by introducing branched siloxane branches”, the interest of introduction of siloxane branches may not be obvious to all readers. I think it is important to explain why using siloxanes to end the peripheral shell, and the obtained dendrons with siloxane ends could be used for which kinds of specific applications, etc. So please offer a more detailed explanation here.
4. The yields of product Lim-G0Cl2, Lim-G1Cl4, Lim-G2Cl8 were not indicated in the “Material and methods”, please add them.
5. Lim-G0Cl2 and Lim-G0Cl4 were prepared by Pt-catalyzed hydrosilylation, how was the Pt catalyst removed? Please give the details in the protocols.
6. Lim-G0Bu2 shown in the Figure 1 is a new compound, but the characteristic data for this compound was not given. Please add it.
7. In the Supplementary document, 13C and 29Si NMRs for compounds Lim-G0Cl2, Lim-G0Cl3, Lim-G1Cl4, Lim-G2Cl8 and Lim-G0.5TMS2, and GPC for Lim-G0Cl2, Lim-G0Cl3, Lim-G1Cl4, Lim-G2Cl8 have not been provided. As authors mentioned in the conclusion of manuscript that “the structures of all compounds were fully characterized by NMR, GPC and elemental analysis”, please complete the analytical data.
8. Some typos need to be corrected.
- Page 2, second paragraph, “dendrimeros” ® “dendrimersomes”
- Page 3 and page 4, “Spayer’s catalyst” ® “Speier’s catalyst”
- Page 7, “Spier’s catalyst” ® “Speier’s catalyst”
Author Response
Reviewer #1:
- A recent review concerning Janus amphiphilic dendrimers (Eur. J. Pharm. Sci., 2017, 97, 113) is worth being cited at the beginning of the Introduction.
Answer: done!
- In this work, the dendron growth center limonene is D-limonene. May I understand that during the synthesis of carbosilane dendrons, the stereo configuration was well retained?
Answer:
The asymmetric carbon atom in the limonene molecule is not involve during the production of dendrons. However, the reactions carried out in this work are not stereospecific, a set of stereomers is formed.
- Page 7, in the last paragraph, “Such a modification of dendrons is interesting, ……, by introducing branched siloxane branches”, the interest of introduction of siloxane branches may not be obvious to all readers. I think it is important to explain why using siloxanes to end the peripheral shell, and the obtained dendrons with siloxane ends could be used for which kinds of specific applications, etc. So please offer a more detailed explanation here.
Answer: «Similar dendrimers are described in literature. For example, in [Milenin, S. A., Selezneva, E. V., Tikhonov, P. A., Vasil’ev, V. G., Buzin, A. I., Balabaev, N. K., Muzafarov, A. M. Hybrid polycarbosilane-siloxane dendrimers: synthesis and properties. Polymers. 2021, 13, 606], carbosilane dendrimers of various generations with heptamethyltrisiloxane substituents in the shell were synthesized. A comparative analysis of the obtained structures with butyl analogues was carried out. It was shown that replacing the butyl shell with heptamethyltrisiloxane does not significantly change the nature of the flow curve, but increases the glass transition temperature of the obtained dendrimers, which is a very useful property of the obtained dendrimers.
Thus, obtaining such systems is a fundamental task. However, you can also get an application orientation. For example, trisiloxane end groups increase gas permeability, this may be a useful property in obtaining gas permeable membranes. Also, trisiloxane tails create a denser shell compared to butyl analogues. This property can be realized in drug delivery systems».
- The yields of product Lim-G0Cl2, Lim-G1Cl4, Lim-G2Cl8 were not indicated in the “Material and methods”, please add them.
Answer: Due to the high reactivity, these compounds have not been isolated in an individual state, therefore, it is impossible to estimate the yield of reaction products.
- Lim-G0Cl2 and Lim-G0Cl4 were prepared by Pt-catalyzed hydrosilylation, how was the Pt catalyst removed? Please give the details in the protocols.
Answer: Due to the high reactivity of these compounds, it was impossible to remove the platinum catalyst at this stage. The platinum complex did not involve with the Grignard reaction. Further, the purification of dendrons from the platinum catalyst was carried out by passing a solution of dendron in toluene through a small layer of silica.
- Lim-G0Bu2 shown in the Figure 1 is a new compound, but the characteristic data for this compound was not given. Please add it.
Answer: done!
- In the Supplementary document, 13C and 29Si NMRs for compounds Lim-G0Cl2, Lim-G0Cl3, Lim-G1Cl4, Lim-G2Cl8 and Lim-G0.5TMS2, and GPC for Lim-G0Cl2, Lim-G0Cl3, Lim-G1Cl4, Lim-G2Cl8 have not been provided. As authors mentioned in the conclusion of manuscript that “the structures of all compounds were fully characterized by NMR, GPC and elemental analysis”, please complete the analytical data.
Answer: Due to the high reactivity, these substances were not isolated in an individual state, therefore, only 1H NMR analysis was performed in order to establish the completeness of the hydrosilylation reaction.
- Some typos need to be corrected.
- Page 2, second paragraph, “dendrimeros” ® “dendrimersomes”
- Page 3 and page 4, “Spayer’s catalyst” ® “Speier’s catalyst”
- Page 7, “Spier’s catalyst” ® “Speier’s catalyst”
Answer: Fixed!
Reviewer 2 Report
The manuscript from Muzafarov and coworkers describes the synthesis of dendrimer cores based upon the diene-containing natural product limonene and containing up to three generations of extension based upon selective hydrosilylation of the less hindered alkene of the diene. Allylation of the chlorosilane intermediate with Grignard reagents generated in situ is followed by capping of the terminal alkenes through addition of alkoxysilanes. Iterations based upon addition of chlorosilanes to the allylated intermediates, followed by another allylation step, accesses higher generation dendrimers. Although there is ample work in the literature on approaches to dendrimer growth, this research builds upon well-established expertise in the group of the corresponding author to explore a new direction based upon the use of a natural product diene in which one of the alkenes can be used as the basis for selective chain growth while the other alkene remains available for further functionalization. The procedures are well-described and the products appear well characterized (more on this below). Overall, the work seems to provide a useful foundation for C/Si-containing dendrimers. Overall, this work will be a useful addition to the literature related to functionalizable/ligatable dendrimrs.
I do have concerns about the manner in which radical thiolation is mentioned yet never tested. For example, in the abstract: “However, the cyclohexene double bond reacts by hydrothiolation.” The discussion has a very similar statement; once again, a reader could easily come to the conclusion that the thiolation had been tested. I understand that the fundamental reactivity pattern is supported by earlier work from the lab of the corresponding author but that result involved a relatively simple core and not a core shrouded by one or more generations of dendrimer. The thiolation should either be tested on some of the intermediates (a great idea, as it would make clear the thiol-based linking could be accomplished within the larger components) or the wording changed to make very clear the authors are referring not to current results but to earlier studies with simpler substrates.
Some areas may require revision. Given that the authors have obtained NMR spectra of many of the intermediates, it is surprising that the spectra are not written up as listings within the experimental section. The elemental analysis results are quite good for the early generation material but are somewhat removed from the theoretical expectation for some of the second-generation products and many of the siloxanes. I am not a polymer chemist so perhaps am asking something not feasible for the field. Would there be value in mass spec characterization of the various species?
The descriptive names assigned to molecules do not always make sense: an example is heptamethyl silyl limonene. I would also suggest showing the structure of hpetamethyltrisiloxane within the Figure 1. I would also recommend closely associating the term “Spayer’s catalyst” with the first mention of the chemical composition “e.g., hexachloroplatinic (IV) hexahydrate” as many readers may be more familiar with an alternate version of the name.
Author Response
Reviewer #2:
- I do have concerns about the manner in which radical thiolation is mentioned yet never tested. For example, in the abstract: “However, the cyclohexene double bond reacts by hydrothiolation.” The discussion has a very similar statement; once again, a reader could easily come to the conclusion that the thiolation had been tested. I understand that the fundamental reactivity pattern is supported by earlier work from the lab of the corresponding author but that result involved a relatively simple core and not a core shrouded by one or more generations of dendrimer. The thiolation should either be tested on some of the intermediates (a great idea, as it would make clear the thiol-based linking could be accomplished within the larger components) or the wording changed to make very clear the authors are referring not to current results but to earlier studies with simpler substrates.
Answer: The main concept of this work is to obtain Janus dendrimers by combining hydrosilylation and hydrothyolation reactions. The hydrothiolation reaction was successfully tested on dendrons of various generations obtained in this article. Now we are publishing only a part of this large work. In the future, the hydrothiolation reaction will be used to connect parts of the Janus-dendrimer.
- Some areas may require revision. Given that the authors have obtained NMR spectra of many of the intermediates, it is surprising that the spectra are not written up as listings within the experimental section. The elemental analysis results are quite good for the early generation material but are somewhat removed from the theoretical expectation for some of the second-generation products and many of the siloxanes. I am not a polymer chemist so perhaps am asking something not feasible for the field. Would there be value in mass spec characterization of the various species?
Answer: During the reactions carried out in this work, a large number of stereomers are formed. Therefore, the description of the NMR spectra as listings will look bulky and uninformative.
- The descriptive names assigned to molecules do not always make sense: an example is heptamethyl silyl limonene. I would also suggest showing the structure of hpetamethyltrisiloxane within the Figure 1. I would also recommend closely associating the term “Spayer’s catalyst” with the first mention of the chemical composition “e.g., hexachloroplatinic (IV) hexahydrate” as many readers may be more familiar with an alternate version of the name.
Answer: done!